# The Medical Triazole Voriconazole Can Select for Tandem Repeat Variations in Azole-Resistant *Aspergillus Fumigatus* Harboring TR_34_/L98H Via Asexual Reproduction

**DOI:** 10.3390/jof6040277

**Published:** 2020-11-11

**Authors:** Jianhua Zhang, Jan Zoll, Tobias Engel, Joost van den Heuvel, Paul E. Verweij, Alfons J. M. Debets

**Affiliations:** 1Laboratory for Genetics, Wageningen University and Research, 6708 PB Wageningen, The Netherlands; joost.vandenheuvel@wur.nl (J.v.d.H.); fons.debets@wur.nl (A.J.M.D.); 2Department of Medical Microbiology, Radboud University Medical Center, 6500 HB Nijmegen, The Netherlands; Jan.Zoll@radboudumc.nl (J.Z.); Tobias.Engel@radboudumc.nl (T.E.); Paul.Verweij@radboudumc.nl (P.E.V.); 3Center of Expertise in Mycology Radboudumc/CWZ, 6500 HB Nijmegen, The Netherlands; 4Center for Infectious Diseases Research, Diagnostics and Laboratory Surveillance, National Institute for Public Health and the Environment (RIVM), 3720 BA Bilthoven, The Netherlands

**Keywords:** azole resistance, tandem repeats variants, asexual reproduction, *Aspergillus fumigatus*, antifungal agents

## Abstract

Azole-resistant *Aspergillus fumigatus* isolates recovered at high frequency from patients, harbor mutations that are associated with variation of promoter length in the *cyp51A* gene. Following the discovery of the TR_34_/L98H genotype, new variations in tandem repeat (TR) length and number of repeats were identified, as well as additional single nucleotide polymorphisms (SNPs) in the *cyp51A* gene, indicating that the diversity of resistance mutations in *A. fumigatus* is likely to continue to increase. Investigating the development routes of TR variants is critical to be able to design preventive interventions. In this study, we tested the potential effects of azole exposure on the selection of TR variations, while allowing haploid *A. fumigatus* to undergo asexual reproduction. Through experimental evolution involving voriconazole (VOR) exposure, an isolate harboring TR_34_^3^/L98H evolved from a clinical TR_34_/L98H ancestor isolate, confirmed by whole genome sequencing. TR_34_^3^/L98H was associated with increased *cyp51A* expression and high VOR and posaconazole MICs, although additional acquired SNPs could also have contributed to the highly azole-resistant phenotype. Exposure to medical azoles was found to select for TR_34_^3^, thus supporting the possibility of in-host selection of TR_34_ variants.

## 1. Introduction

In recent decades, the emergence of azole resistance in *Aspergillus fumigatus*, a saprobic filamentous fungus, has caused challenges as this fungus is a main cause of opportunistic fungal diseases in humans and animals. In the past five years, the azole resistance frequency increased from 7.6% in 2013 to 15% in 2018 in the Netherlands [1]. Up to 90% of resistant *A. fumigatus* isolates cultured from patient samples exhibit a resistance mechanism characterized by tandem repeats (TR) in the promotor region of the *cyp51A* gene in combination with short nucleotide polymorphisms (SNPs) in the gene, typically TR_34_/L98H and TR_46_/Y121F/T289A. *Cyp51A* encodes the target enzyme 14α-demethylase in the cell membrane and these mutations confer resistance to triazole antifungals including itraconazole (ITR), voriconazole (VOR), isavuconazole (ISA) and posaconazole (POS) [2,3,4,5]. In addition to clinical cultures, identical TR-associated resistance mutations were recovered from environmental samples, especially in decaying plant material that contained azole fungicide residues [6]. Numerous studies have confirmed a relation between environmental resistance selection and azole-resistant infection in humans, and the concept of an environmental route of resistance selection is now generally accepted [7,8]. TR-mediated resistance mutations could be distinguished from single resistance SNPs that were commonly found in resistant *A. fumigatus* isolates from patients that received chronic azole therapy (patient route). However, the association between TR-mediated resistance mutations and resistance selection in the environment was recently challenged when an azole-resistant *A. fumigatus* isolate harboring TR_120_ was recovered from a patient that had received chronic azole therapy [5]. Whole genome sequencing of a previously cultured wild type (WT) *A. fumigatus* isolate and the TR_120_ isolate proved that the TR resistance mechanism had emerged through patient therapy. Recently a TR_34_^3^/L98H variant was recovered from a cystic fibrosis patient who also harbored a TR_34_/L98H isolate. Microsatellite typing showed that the TR_34_^3^/L98H and TR_34_/L98H shared eight of nine markers, which supports possible in vivo acquisition of an additional TR [9]. Apparently not the resistance selection route, but other factors such as chemical characteristics of the azole, the reproduction mode the fungus goes through and azole stress duration might determine which resistance mutation is selected for [10].

It is important to understand how TR-mediated resistance mutations develop. TR-mediated resistance is highly dominant, especially TR_34_, accounting for up to 70% of resistance mutations in surveillance studies [11]. Furthermore, over time increased variation in TRs is observed with both increasing TR length, i.e., TR_34_, TR_46_, TR_53_, and TR_120_, as well as increasing numbers of TR-duplications, i.e., TR_46_, TR_46_^3^ and TR_46_^4^ [5,12,13] and TR_34_ and TR_34_^3^ [9].The repeated promoter sequence in the variants TR_46_, TR_53_, TR_120_ TR_46_^3^, and TR_46_^4^ all contain the repeated sequence of TR_34_. It has been shown that the repeat sequence in TR_34_ is bound by both the sterol regulatory element binding protein SrbA, and the CCAAT binding complex (CBC). Sterol biosynthesis and azole tolerance is governed by the opposing actions of SrbA and the CCAAT Binding. The mechanism underpinning TR_34_-driven overexpression of *cyp51A* is a result of the duplication of SrbA binding sites but not CBC [14]. Therefore, the extra two copies of 34 bp, largely increase the overexpression of *cyp51A* due to the extended SrbA binding sites.

In a previous study we have shown that sexual reproduction can play a role in extending the promoter length under laboratory conditions by unequal crossing over between tandem repeats during meiosis. Through sexual crossing of two *A. fumigatus* isolates harboring TR_46_, an isolate with TR_46_^3^ was recovered in the progeny [4]. However, sexual reproduction in *A. fumigatus* has not yet been found in the environment and may be restricted to highly specific conditions. Furthermore, the sexual cycle requires compatible strains of opposite mating type and is highly unlikely to occur in the clinical setting. It is unclear whether TR diversification can occur in asexually reproducing haploid cultures. Our hypothesis is that also asexual reproduction can extend the promoter length, since asexual reproduction involves numerous mitotic divisions and promoter repeats may further expand by replication slippage or unequal sister chromatid recombination [15,16]. Additionally, as TR-mediated resistance was shown to emerge in an azole-treated patient we investigated if exposure to VOR could play a role in TR-variations through the asexual cycle.

In this study, we performed experimental evolution which allows for asexual reproduction to occur using a clinical *A. fumigatus* TR_34_/L98H isolate and exposure to VOR.

## 2. Materials and Methods

Culture media and antifungal agents: Malt extract agar (MEA), used for counting conidia and measuring the mycelial growth rate, was purchased from Sigma Company (Sigma, Aldrich, Germany). Minimal Medium (MM) was used for culturing the heterokaryon. MM consists of 6.0 g NaNO_3_, 1.5 g KH_2_PO_4_, 0.5 g MgSO_4_. 7H_2_O, 0.5 g KCl, 10 mg of FeSO_4_, ZnSO_4_, MnCl_2_, and CuSO_4_ and 15 g agar + 1000 mL H_2_O (pH 5.8) [17]. The antifungal agents ITR, VOR and POS were purchased from Sigma Company (Sigma Aldrich, Germany) [18].

The evolution of resistance through asexual reproduction under antifungal agents pressure: we started the evolutionary experiment with clinical isolate TR_34_/L98H (V30-40) as ancestral strain under exposure to 2 µg/mL of VOR, which allows the asexual sporulation of ancestor TR_34_/L98H (V30-40). In the evolutionary experiment, a new cycle was started every seven days. This evolutionary line was inoculated by placing a droplet (5 µL) of asexual conidia suspension from the ancestor colony into a 30 mL glass bottle (infusion bottle, VWR, Netherlands) with MEA supplemented with VOR and the bottle was incubated at 37 °C. The colonies grown in these bottles represented the first cycle. After seven days, *A. fumigatus* conidia were harvested with 5 mL saline (distilled water with NaCl 0.8 g/L) supplemented with Tween 80 (0.05%), by washing off the fungal growth with beads. Asexual conidia suspensions were stored and an inoculum of 1% (50 µL) of the conidia was used to initiate a next selection cycle [18]. This procedure was repeated every seven days until 15 cycles were completed.

The analysis of mutations (promoter length and coding region) of the *cyp51A* gene: at the end of the evolutionary experiment, the whole population was diluted and plated on agar plates supplemented with 2 µg/mL of VOR. After two days of growth under 37 °C, 20 *A. fumigatus* colonies were randomly selected. Single resistant colonies were inoculated in 3 mL Malt Extract broth and grown overnight at 37 °C. Mycelial mats were recovered and subjected to a DNA extraction protocol [19]. The full coding sequence of the *cyp51A* gene and its promoter region were determined by amplification and subsequent sequencing as previously described [20,21]. The *cyp51A* sequence under accession number AF338659 in GenBank was used for comparison to detect mutations, *cyp51A*-PF(5′-ATGGTGCCGATGCTATGG-3′) and *cyp51A*-PR (5′-CTGTCTCACTTGGATGTG-3′). The presence of TRs in the promotor region of the gene was investigated by amplifying part of the promoter region of the *cyp51A* gene using appropriate primers (5′-TGAGTTAGGGTGTATGGTATGCTGGA-3′ and 5′-AGCAAGGGAGAAGGAAAGAAGCACT-3′). The cycling program consisted of a 3-min denaturation step at 94 °C, followed by 35 cycles of 30 s at 94 °C, 45 s at 55 °C, and 45 s at 72 °C and a final elongation step of 5 min at 72 °C [13]. The amplified DNA fragments were purified with a QIAquick PCR purification kit (Qiagen). DNA sequencing of the forward strand of each fragment was performed by Eurofins sequencing company (Ebersberg, Germany). The resulting sequences were aligned in CLUSTALW46 using the program BioEdit47. If genetics changes were detected, the early evolutionary cycles were traced back following the same protocol.

Testing for isogeneity between ancestor and evolved isolates: In order to test whether the evolved *A. fumigatus* colony was isogenic with the ancestor strain, we applied heterokaryon compatibility testing, microsatellite locus genotyping, and whole genome sequencing (WGS).

Heterokaryon compatibility test: as a result of high polymorphism for heterokaryon incompatibility het-genes involved in self/nonself recognition in fungal populations, heterokaryon compatibility is mostly restricted to clonally related isolates and, therefore two randomly picked isolates from nature are typically heterokaryon incompatible [22,23]. In this study heterokaryon compatibility of strains was tested following standard methods [24,25,26]. Recessive markers (nitrate non-utilizing mutations nia and cnx) were introduced by ultra violet (UV 60 s; 20 erg/mm^2^/s) radiation in ancestor isolate TR_34_/L98H and any evolved genotypes. Nia and cnx mutants were isolated on basis of chlorate resistance and characterized for their ability to utilize urea, hypoxanthine, nitrite and nitrate as sole N-source (a *nia* (chlorate resistant, nitrate non-utilising, hypoxanthine and nitrite utilising) and a *cnx* (chlorate resistant, nitrate and hypoxanthine non-utilising, nitrite utilising), see Appendix A). The isolates with complementing nia and cnx markers were co-inoculated on the medium with nitrate as the sole N-source. Heterokaryons of compatible complementing strains show typical vigorous growth after 5 days, whereas inability to form heterokaryons results in spares growth of the individual homokaryons [27].

Microsatellite locus genotyping: genetic relationships between the ancestor and evolved genotype were established based on a categorical analysis of six microsatellite markers (STR3 A, B, C and STR4 A, B, C) [13].

Whole genome sequencing: for ancestral TR_34_/L98H and evolved genotypes we performed WGS. Environmental isolate TR_34_/L98H (V62-10) was used as control.

Mapping and variant calling: Raw FASTQ files were trimmed and filtered by TRIMMOMATIC (v 0.27, LEADING:3, TRAILING:3, SLIDINGWINDOW:4:15, MINLENGTH:70) [28]. The resulting trimmed reads were aligned using bwa mem (v 0-7-15, default parameters) [29] using the *A. fumigatus* Af293 as reference (assembly ASM265v1). Reads with mapping quality lower than 20 were filtered using samtools (v 0.1.19) [30]. We then removed duplicates using picard tools (v.1.109, http://broadinstitute.github.io/picard/) and performed realignment around indels with GATK (v 3.7-0) [31].

The resulting BAM files were combined in mpileup format (samtools, mpileup, default parameters, v 0.1.19) [30], after which SNPs and indels were separately called using varscan (v 2.3.9, using mpileup2snps and mpileup2indel, respectively, -output-vcf 1, -min-var-freq 0.05) [32]. The resulting vcf files were inspected for genetic differences between any pairwise comparison, using vcfR (v 1.8.0) [33], for which we used a minimum allele frequency difference of 0.6 and a minimum coverage of 10 to filter SNPs and indels. Every resulting variant was then visually inspected in all samples using IGV (v 2.4.14) [34].

Susceptibility testing and *cyp51A* gene expression of ancestor and evolved isolates: In vitro susceptibility testing was performed using a broth microdilution method according to EUCAST protocol E.DEF 60 9.2 [13,18]. The *cyp51A* expression was analyzed from duplicate cultures of the ancestor TR_34_/L98H isolate and evolved genotypes and a WT control isolate (V256-07). Strains were cultured for 16 h in 50 mL of Vogel’s MM at 37 °C at 200 rpm. Harvested mycelia were snap-frozen in liquid nitrogen and homogenized with a MagNALyser. RNA was isolated using RNAzolB (Sigma, Aldrich, Germany) according the manufacturers protocol. cDNA was synthesized using random hexamer primers and Transcriptor reverse transcriptase (Roche, Meylan, France). Real-time PCR was performed for the *cyp51A* and actin genes using the PCR Master kit (Roche, Meylan, France). *Cyp51A* and actin amplification was performed using specific primer/probe sets for each gene (*cyp51A*: forward primer 5′- GTGCTCCTTGCTTCACCTG-3′, reverse primer 5′- TCCTGCTCCTTAGTAGCCTGGTT-3′, probe 5′-6Fam-AGTGACAGCCCTCAGCGACGAA-BBQ-3′; Actin: forward primer 5′-ATTGCTCCTCCTGAGCGTAAATAC-3′, reverse primer 5′-GAAGGACCGCTCTCGTCGTAC-3′, probe 5′-6Fam-TCTGGCCTCTCTGTCCACCTTCCA-BBQ-3′). Expression levels were calculated using the delta–delta–Ct method and normalized for WT expression levels [35].

The mycelium growth rate and conidia production of ancestor and evolved genotype: the mycelial growth rate (MGR) and conidia production of ancestor TR_34_/L98H and evolved genotypes were assayed both in the presence (2 µg/mL) and absence of VOR. We inoculated 5 µL of conidia suspension in a 30 mL glass bottle of MEA medium with and without 2 μg/mL of VOR. After four days of incubation at 37 °C, conidia were harvested with 5 mL saline (distilled water with NaCl 0.8 g/L) supplemented with Tween 80 (0.05%), by washing off the fungal conidia with beads. The total conidia production was measured in three replicates by a Casy^®^ TT cell counter (OLS OMNI Life Science, Germany). For MGR determination, we inoculated 2 µL of the conidia suspension of all cultures onto Petri dishes with solid MEA medium with and without 2 μg/mL of VOR. After four days of incubation at 37 °C, the MGR was determined by averaging the colony diameters (in mm) as measured in two randomly chosen perpendicular directions.

Competition between the ancestor and evolved isolate: In order to investigate the competition between the ancestor TR_34_/L98H and evolved genotypes, we visually selected a spontaneous white color mutant from the ancestral TR_34_/L98H by microscopic inspection of a three-day-old culture grown on an MEA plate covered with a high-density of conidial sporeheads. In order to check whether the color has effect on VOR susceptibility and competition with the other isolate, TR_34_/L98H green conidia and white conidia were mixed (ratio 1:1) and co-inoculated in the center of an agar-plate in the presence and absence of VOR. After two and half days of growth, the distribution of white and green conidia was recorded via camera. After the confirmation of the color effect, a mixed inoculum of conidia from the white ancestral TR_34_/L98H strain and the green evolved genotypes (ratio 1:1) was plated on MEA-plates in the presence and absence of 2 µg/mL VOR. After two days of growth, the distribution of white and green conidia was recorded in the same way.

## 3. Results

### 3.1. Evolution of Azole Resistance

At the end of the 15-week evolutionary experiment, 20 single conidia cultures of this line were investigated for changes in the promoter region regarding length and coding region of *cyp51A* gene by PCR. The sequence results showed that 100% (20/20 colonies) had genetic changes in the *cyp51A* gene. All isolates contained three copies of 34 bp in the promoter region and a point mutation L98H on the coding region. Thus, compared with the ancestral isolate an additional copy of 34 bp had been gained. The previous evolutionary cycles were then analyzed, which showed that the genetic change first emerged in the fifth cycle with a percentage of 5%. By the eight cycle, the proportion of TR_34_^3^/L98H harboring isolates had increased to 50%, while 100% of the colonies harbored TR_34_^3^/L98H at the 12th cycle (Figure 1).

### 3.2. The Ancestor TR_34_/L98H and Evolved TR_34_^3^/L98H Are Isogenic

To exclude contamination, we used the self/nonself compatibility test and microsatellite locus genotyping methods to assess whether the evolved isolate and ancestral isolate are isogenic. Both genotypes were found to be compatible (Figure 2) and isogenic (Table 1).

WGS was performed on the ancestor TR_34_/L98H, the TR_34_^3^/L98H, and an environmental TR_34_/L98H control isolate (raw data are available via NCBI, BioProjectID: PRJNA666755). TR_34_/L98H and TR_34_^3^/L98H were confirmed to be isogenic, sharing 99.9% of their genomes. Only three SNPs were found to differ between ancestral TR_34_/L98H and TR_34_^3^/L98H, while 14,500 SNPs were found between TR_34_^3^/L98H and the environmental TR_34_/L98H control isolate (Appendix A). These three SNPs included one in gene AFUA_1G06920 Serine/threonine protein kinase which may be involved in growth (http://www.aspergillusgenome.org/cgi-bin/locus.pl?locus=AFUA_1G06920&organism=A_fumigatus_Af293); and an AFUA_5G06450/Vacuolar protein sorting protein DigA which may be involved in vacuole organization (http://www.aspergillusgenome.org/cgi-bin/locus.pl?locus=AFUA_5G06450&organism=A_fumigatus_Af293); The third gene involved AtrR, a transcription factor that has been reported to play a role in regulating *cyp51A* expression [36].

### 3.3. Susceptibility Testing and Cyp51a Gene Expression of Ancestor and Evolved Isolates

In vitro susceptibility testing showed increased MICs of VOR and POS in the TR_34_^3^/L98H isolate (Table 2). Expression levels assayed from Cyp51A/actin mRNA ratios showed that *cyp51A* gene expression levels in the TR_34_^3^/L98H were significantly higher compared to WT and ancestral TR_34_/L98H (ANOVA: *p* < 0.05) (Figure 3).

### 3.4. The Role of SNPs in the Observed Azole-Resistant Phenotypes

As the AtrR gene is involved in regulation of cyp51A expression, mutations in this gene might be associated with an azole-resistant phenotype. We therefore analyzed an isolate from the eighth cycle, which harbored the TR_34_/L98H mutation, but showed a change in VOR phenotype (MIC >16 µg/mL) and POS (MIC > 8 µg/mL) compared with the ancestor TR_34_/L98H isolate. This isolate, indicated as Transient TR_34_/L98H, contained a mutation in AtrR, which supports a possible role of AtrR in azole resistance (Appendix A). This isolate also harbored an additional non-resistant related SNP in protein PtaB, which was not found in the finally evolved TR_34_^3^/L98H isolate, indicating that this line did not survive in competition with the TR_34_^3^/L98H line (Appendix A). Cyp51A expression level of the Transient TR_34_/L98H isolate was higher than that of the ancestor TR_34_/L98H, but lower than the expression level of TR_34_^3^/L98H (Appendix A). These observations indicate that in the evolved TR_34_^3^/L98H, the overexpression of cyp51A is the combined effect of AtrR and three copies of 34 bp (see Appendix A). Furthermore, the fact that in our evolution experiment TR_34_^3^/L98H replaced Transient TR_34_/L98H indicates a selective advantage of TR_34_^3^/L98H over Transient TR_34_/L98H under VOR pressure.

### 3.5. The Mycelium Growth Rate and Conidia Production of Ancestor and Evolved TR_34_ Genotypes

Under VOR (2 µg/mL) exposure, TR_34_^3^/L98H showed 60% increased mycelium growth rate and 2.6 times the conidia production compared with the ancestral TR_34_/L98H (ANOVA: *p* < 0.05, Figure 4). However, in the absence of VOR the TR_34_^3^/L98H isolate showed a fitness cost with lower amounts of conidia production (30% less) and decreased mycelium growth rate compared to the ancestral TR_34_/L98H (Figure 4).

### 3.6. Competition between TR_34_/L98H and TR_34_^3^/L98H

Fitness differences between TR_34_/L98H and TR_34_^3^/L98H in presence or absence of VOR can be visualized when conidia are co-inoculated in the center of an agar-plate. As a control, TR_34_/L98H with green conidia and a derived mutant with white conidia were mixed. They showed the same fitness with equal appearance of white and green conidia at the rim of the colony in the presence and absence of VOR after 2.5 days of growth at 37 °C (Figure 5B,D). This indicates that there is no color effect on fitness of the white-color mutation. Under VOR-free conditions, TR_34_^3^/L98H with green conidia was clearly outcompeted by the faster growing TR_34_/L98H with white conidia as the rim has exclusively white conidia (Figure 5A). However, in the presence of VOR (2 µg/mL), TR_34_^3^/L98H outcompeted the ancestor TR_34_/L98H isolate, showing only green conidia at the rim but no white conidia (Figure 5C) after two days.

## 4. Discussion

Our results indicate that *cyp51A* promoter elongation in *A. fumigatus* is possible via asexual reproduction when exposed to VOR. After five evolutionary cycles the TR_34_^3^/L98H genotype first emerged, and subsequently became the dominant genotype with higher VOR and POS MICs than the ancestor. The extra 34 bp promoter repeat resulted in a 1.5 times higher expression of *cyp51A* compared with the TR_34_/L98H ancestor isolate. TR_34_^3^/L98H manifested better fitness compared with the ancestor TR_34_/L98H when grown in the presence of VOR, in terms of mycelium growth rate and conidia production. These results indicate that under strong azole selection the TR copy number may increase through asexual reproduction in *A. fumigatus*. Thus *A. fumigatus* may employ both the sexual and asexual cycle to expand promotor repeats and adapt to the azole environment [4].

Mechanistically this can be explained by replication slippage or unequal sister chromatid recombination during mitosis [37]. Such processes may be rare per single mitotic division but are relevant during growth and sporulation of *A. fumigatus* cultures that involve numerous mitotic divisions. Typically, after one week of growth, a colony of *A. fumigatus* started from a single conidium may contain more than 10^9^ conidia. Apart from nuclear divisions leading to multinucleate mycelium, during sporulation each conidium results from a mitotic division in a phialide cell of a conidiophore. During each of these numerous mitotic divisions, there is a possibility of replication slippage and unequal sister chromatid recombination that may lead to an increase in the TR number (Figure 6).

Replication slippage may occur during DNA replication in the mitotic cell cycle. Particularly during the S stage of interphase (from N-2N, Figure 6A), when DNA is replicated. Local misalignment of the template and nascent strands during a brief detachment of the DNA polymerase may lead to slippage and the TRs obtained are expanded [38,39] (Figure 6A). After the replication stage, two sister chromatids are generated and at S/G2 phase unequal cross over may occur between the TRs of two sister chromatids, resulting in a daughter cell with three copies of 34 bp after chromatid segregation [40] (Figure 6B).

Our study has shown the acquisition of an extra TR in a clinical isolate, which is remarkably different from a previous study, in which TR_34_^3^ was recovered through exposure of recombinant isolate *aku*B^KU80^ to tebuconazole under laboratory conditions [7]. Recombinant isolate *aku*B^KU80^ had an increased mutation frequency due to a defect in the DNA repair which may have facilitated TR triplication. Our results indicate that in-host selection of azole-resistant TR_34_^3^
*A. fumigatus* isolates is possible and may further limit treatment options. Our laboratory observations support the recent report of recovery of genetically closely related TR_34_^3^/L98H and TR_34_/L98H isolates from a cystic fibrosis patient. The patient had been treated with VOR and POS (MC Arendrup, personal communication), which could have selected TR_34_^3^/L98H harboring *A. fumigatus* strains in the lung [9]. However, the TR_34_/L98H patient isolate was recovered after the isolates harboring TR_34_^3^/L98H and no ancestral isolate was available [9], therefore it is unclear whether the TR_34_^3^-variant evolved in the host under VOR and POS selection from a TR_34_ or vice versa. Our study shows that the TR_34_^3^-variant can evolve from a TR_34_ ancestor through asexual reproduction. The TR_34_^3^/L98H patient isolates showed no phenotypic change regarding VOR and POS MICs compared to the TR_34_/L98H patient isolate [9]. However, the TR_34_^3^/L98H isolate we obtained through exposure to VOR showed higher VOR and POS MICs compared with the ancestor. The TR_34_^3^/L98H isolate harbored three SNPs compared with the TR_34_/L98H ancestor, one of which (AtrR) was found to be a transcription factor reported to play a role in regulating *cyp51A* expression. This mutation has been confirmed to contribute to the high VOR and POS resistance phenotype as analysis of an isolate obtained from the 8^th^ cycle showed high resistance (VOR MIC > 16 µg/mL and POS MIC >8 µg/mL) in the TR_34_/L98H background. The *cyp51A* expression level of this isolate was higher compared with the TR_34_/L98H ancestor (Appendix A), but lower than final fixed TR_34_^3^/L98H, which might have contributed to a fitness benefit of TR_34_^3^/L98H compared to the Transient TR_34_/L98H under VOR selection pressure. Understanding the role of AtrR in the VOR resistance phenotype was beyond the scope of our research, and further experiments would be required. As antifungal drug and other stressor exposures in the lung are uncontrolled [41], whole genome sequencing would be required to gain more insights into the genetic background of the patient isolates in relation to the observed triazole phenotypes. Our observations are in line with observations of in-host selection of triazole resistance mutations, including TR_120_ from an ancestor isolate without a TR [5]. Nevertheless, despite TR_34_/L98H being the most prevalent azole-resistance mechanism, an isolate with TR_34_^3^ has been reported only once indicating that specific conditions might be required for its selection. Isolates with three copies (TR_46_^3^) and four copies (TR_46_^4^) have been found and acquisition of an extra TR was demonstrated in two environment isolates carrying TR_46_/Y121F/T289A through unequal crossover during sexual reproduction [4].

There are various possible reasons for the low frequency of TR_34_^3^ isolates. There may be a fitness cost of TR_34_^3^ isolates in the azole-free environment, which was also observed in the isolate that we analyzed. The consequence of a fitness cost is that the genotype will disappear in competition with WT isolates unless it acquires compensatory mutations that restore its fitness. In our experiment, TR_34_
^3^/L98H is likely not to have undergone compensatory evolution to overcome its fitness cost in the azole-free environment. Alternatively, other factors related to azole selection pressure (type of azole, concentration, duration) may be important. Furthermore, TR_34_^3^ may not be stable in the long term.

Although TR-mediated azole-resistance mutations are generally associated with environmental resistance selection and single resistance mutations with in-host resistance selection, there is increasing awareness that the correlation between resistance-mutation characteristics and route of selection is artificial [10]. Indeed, *A. fumigatus* harboring single resistance mutations has been recovered from the environment and from azole-naïve patients, which is consistent with environmental resistance rather than in-host selection [10]. Alternatively, TR_120_ was shown to develop through in-host selection [5]. Our observation of TR_34_^3^ selection through exposure to a medical triazole underscores that characteristics of the fungus and the azole environment are the critical factors that determine which genotype will eventually evolve.

In this study, we provide evidence for selection of TR-variants in TR_34_ isolates through asexual reproduction during VOR exposure, adding to the strategies *A. fumigatus* can employ to adapt to the azole environment. Future work should further characterize factors that facilitate selection of TR-variants including genetic background of the isolate, concentration and chemical properties of the azole in order to enable development of preventive interventions.

## Figures and Tables

**Figure 1 jof-06-00277-f001:**
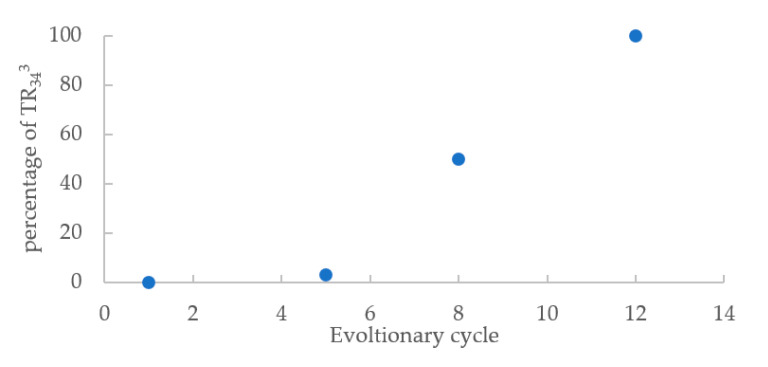
The percentage of isolates harboring TR_34_^3^/L98H over the evolutionary cycle.

**Figure 2 jof-06-00277-f002:**
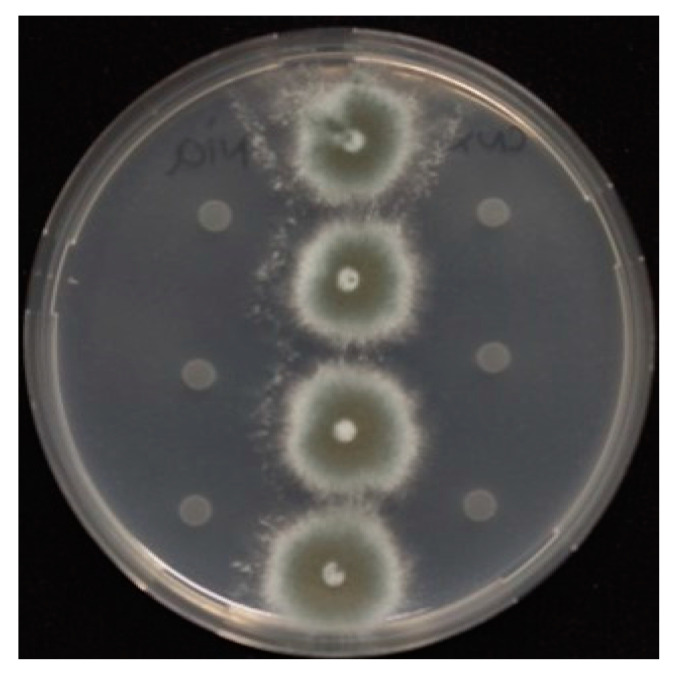
Heterokaryon compatibility test on agar with nitrate as the sole N-source. Left: ancestor TR_34_/L98H nia mutant, right: evolved TR_34_^3^/L98H cnx mutant. Middle, the mixture of the TR_34_/L98H nia and TR_34_^3^/L98H cnx mutants from left and right. The figure shows that the strains were heterokaryon compatible: the strains are not able to grow individually on nitrate as a N-source but can grow together through complementation in a heterokaryon. From top to bottom, 4 replicates are shown.

**Figure 3 jof-06-00277-f003:**
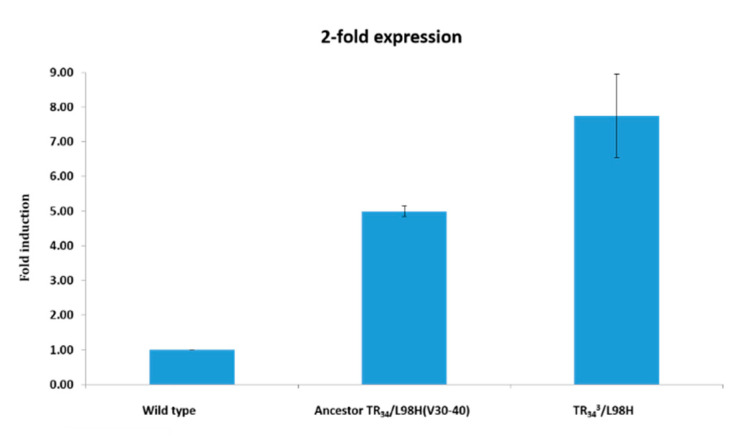
Expression of *cyp51A* in the TR_34_^3^/L98H isolate, compared with the TR_34_/L98H ancestor and a wild type (WT) control isolate.

**Figure 4 jof-06-00277-f004:**
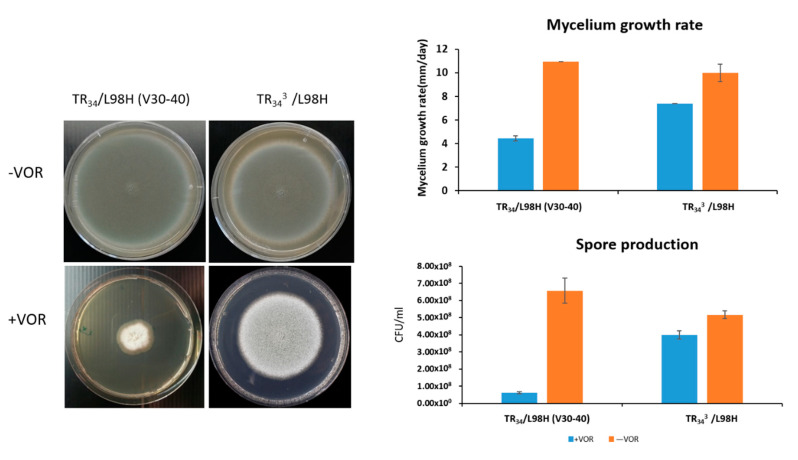
The growth characteristics of TR_34_/L98H and TR_34_^3^/L98H, including the phenotype, growth rate and conidia production when cultured in the presence and absence of voriconazole (VOR) (2 µg/mL).

**Figure 5 jof-06-00277-f005:**
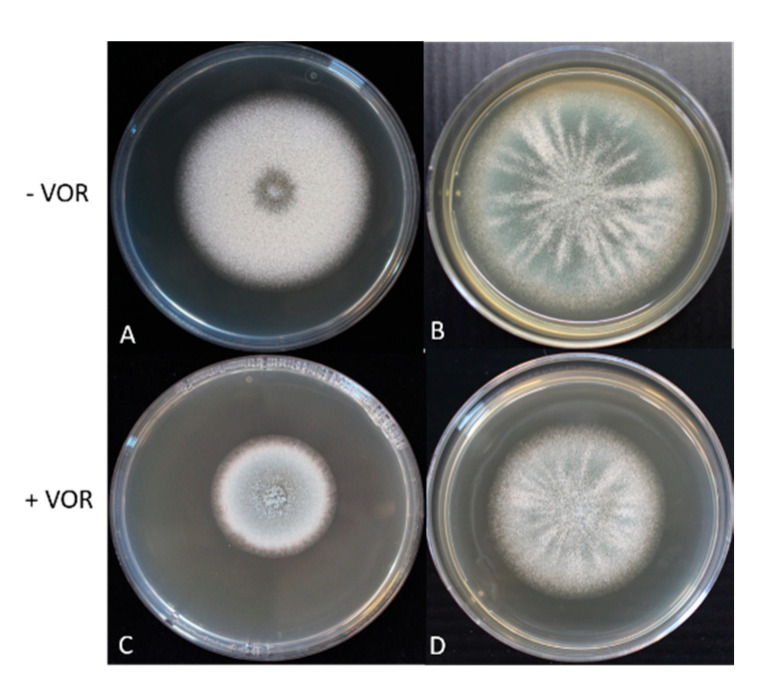
Competition between TR_34_/L98H (white conidia) and TR_34_^3^/L98H (green conidia) in the absence (panel **A**) and presence (panel **C**) of VOR. Control between TR_34_^3^/L98H (green conidia) and TR_34_^3^/L98H (white conidia) in the absence (panel **B**) and presence (panel **D**) of VOR after two and half days of growth at 37 °C.

**Figure 6 jof-06-00277-f006:**
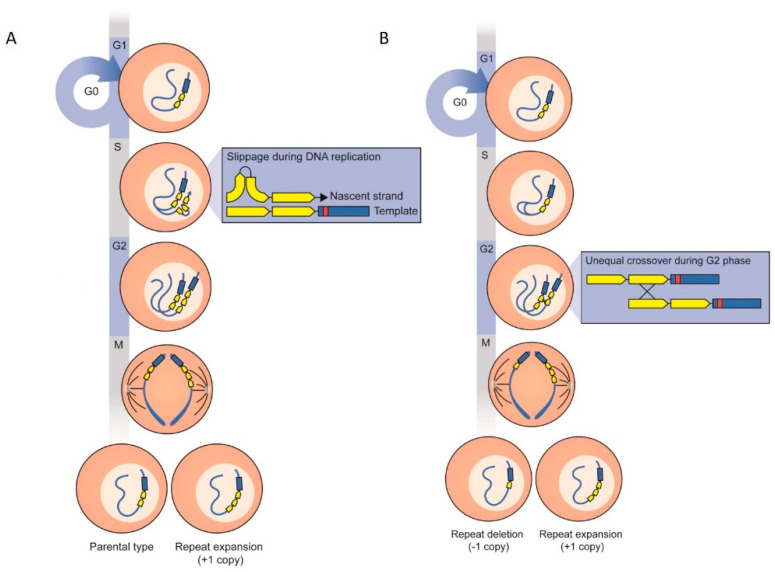
Possible mechanisms of extension from two to three 34 bp tandem repeats (TRs) in the promoter region of the *cyp51A* gene during the mitotic cell cycle. (**A**) Replication slippage: when strand slippage occurs during DNA replication in the S-phase of the mitotic cell cycle, a DNA strand with the repeat may loop out resulting in addition of the repeat number [15]. (**B**) Unequal sister chromatid recombination: during the S/G2 phase, unequal crossing over may occur between the repeats of the sister chromatids resulting in an extra copy of the repeat on one chromatid.

**Table 1 jof-06-00277-t001:** Microsatellite locus genotyping.

		STR 3 *			STR 4 *		
Type	Number	3A	3B	3C	4A	4B	4C
Ancestor	TR_34_/L98H	31	9	8	6	8	19
Evolved	TR_34_^3^/L98H	31	9	8	6	8	19

* STR3 A, B, C and STR4 A, B, C: six microsatellite markers.

**Table 2 jof-06-00277-t002:** In vitro azole susceptibility profiles of TR_34_/L98H and TR_34_^3^/L98H.

			MIC (µg/mL)	
Isolate Code	Genotype	ITR	VOR	POS
**V30–40**	TR_34_/L98H	>16	4	0, 5
**v284–37**	TR_34_^3^/L98H	>16	>16	>8

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
