# Peer review of "The Medical Triazole Voriconazole Can Select for Tandem Repeat Variations in Azole-Resistant Aspergillus Fumigatus Harboring TR34/L98H Via Asexual Reproduction"

_jof, 2020, doi:10.3390/jof6040277_

Round 1
Reviewer 1 Report
Authors in the manuscript “The medical triazole voriconazole can select for tandem repeat variations in azole-resistant Aspergillus fumigatus harboring TR34/L98H via asexual reproduction” presented data from experimental work of azole-resistant Aspergillus fumigatus harboring TR34/L98H via asexual reproduction. Authors presented very interesting and important information, which could provide additional insights in occurrence of azole-resistant A. fumigatus after therapy with azoles.
I have couple specific questions, which I believe will improve manuscript for readership. Please see below:
- Line 92: Please check, if this Epsom salt, should formula be: MgSO4·7H2O?
- Line 92: missing coma: “…MnCl2, and…”
- Line 143-146: Sentence little confusing. Please check if it is missing or redundant?
- Line 149: Here and in multiple places seems that parenthesis are open, but not closed. Please check through the manuscript.
- Line 183: Figure 1 legend. From text it seems that this is expressed the percentage of isolates harboring TR343/L98H. Would be good to make clear in the legend, that figure could stay by itself.
- Line 212: Did authors looked at mycelium growth and spore production of WT control isolate? It might be interesting for comparison.
- Line 214: Do authors have explanation for fitness loss as stated in absence of VOR? Is this a result of exposure to VOR during the experiment?
- Line 240-241: is better growth of TR343/L98H in presence of VOR result of resistance and higher MIC? Are these isolates not growing because VOR does not have effect, or that they become more fit?
Author Response
- Line 92: Please check, if this Epsom salt, should formula be: MgSO4·7H2O?
- Line 92: missing coma: “…MnCl2, and…”
Reply****: Thank you for your suggestions, we have changed in L92.
- Line 143-146: Sentence little confusing. Please check if it is missing or redundant?
Reply****: Thank you for your comment, we have adjusted in the main text as below.
Mapping and variant calling: Raw FASTQ files were trimmed and filtered by TRIMMOMATIC (v 0.27, LEADING:3, TRAILING:3, SLIDINGWINDOW:4:15, MINLENGTH:70)[28]. The resulting trimmed reads were aligned using bwa mem (v 0-7-15, default parameters) [29] using the A. fumigatus Af293 as reference (assembly ASM265v1).
- Line 149: Here and in multiple places seems that parenthesis are open, but not closed. Please check through the manuscript.
Reply****: Thank you for your comment, we have adjusted in the main text as below.
The resulting BAM files were combined in mpileup format (samtools, mpileup, default parameters, v 0.1.19) [30], after which SNPs and indels were separately called using varscan (v 2.3.9, using mpileup2snps and mpileup2indel resp., --output-vcf 1, --min-var-freq 0.05) [32]. The manuscript was checked throughout for inconsistent parenthesis.
- Line 183: Figure 1 legend. From text it seems that this is expressed the percentage of isolates harboring TR343/L98H. Would be good to make clear in the legend, that figure could stay by itself.
Reply****: We have changed in L205.
- Line 212: Did authors looked at mycelium growth and spore production of WT control isolate? It might be interesting for comparison.
Reply****: Thank you for your thoughtful review. Since the control isolate has a WT phenotype, it does not grow on VOR (2µg/ml) and has very limited conidia production. Therefore, including mycelium growth and conidia production of WT we believe does not add relevant information. We propose not to include these data.
- Line 214: Do authors have explanation for fitness loss as stated in absence of VOR? Is this a result of exposure to VOR during the experiment?
Reply****: A fitness cost is common in isolates that acquire resistance mutations. In Discussion section L327-332, we discuss the fitness loss of TR343/L98H. “a fitness cost of the TR343 isolate in the azole-free environment was observed. The explanation of this cost is the TR34 3/L98H is likely not to have undergone compensatory evolution to overcome its fitness cost in the azole-free environment.”
- Line 240-241: is better growth of TR343/L98H in presence of VOR result of resistance and higher MIC? Are these isolates not growing because VOR does not have effect, or that they become more fit?
Reply****: Thank you for your comments. Because these isolates are VOR resistant, the presence of VOR does not cause a stress response based on gene expression levels (data not shown), and the growth rate of the fungus not affected. No change to the text was made.

Reviewer 2 Report
The authors of “The medical triazole voriconazole can select for tandem repeat
variations in azole-resistant Aspergillus fumigatus harboring TR34/L98H via
asexual reproduction” provide information regarding the association of azole
exposure on the selection of TR variations during the asexual reproduction of
A. fumigatus. It is an interesting topic, but the paper is too long and difficult
to read. Some specific comments are given below
Lines 89, 98, spores or conidia and elsewhere where "asexual spores" were
mentioned? The use of conidia vs spores needs to be clarified. Usually,
conidia are the asexual elements and spores the sexual. Not sure what using the term clinical before antifungals; antifungal agents is
most commonly used. What were the original MICs of the strain(s) used? The subtitles such as “Heterokaryon compatibility test:” should be made more
prominent by larger font and/ or bolding? Or similar to what is was done under
“Results”. What was the MIC data for the strains used prior to the experiment,
they should be provided in Table 2 ; after all, the term WT is used. Which
bring us to what was the basis for the use of the 2 mcg/ml?
Author Response
Response to Reviewers 2
The authors of “The medical triazole voriconazole can select for tandem repeat
variations in azole-resistant Aspergillus fumigatus harboring TR34/L98H via
asexual reproduction” provide information regarding the association of azole
exposure on the selection of TR variations during the asexual reproduction of
A. fumigatus. It is an interesting topic, but the paper is too long and difficult
to read. Some specific comments are given below.
- Lines 89, 98, spores or conidia and elsewhere where "asexual spores" were
mentioned? The use of conidia vs spores needs to be clarified. Usually,
conidia are the asexual elements and spores the sexual.
Reply****: We have adjusted our words in main text from spores to conidia.
- Not sure what using the term clinical before antifungal agents; antifungal agents is
most commonly used. What were the original MICs of the strain(s) used?
Reply****: We intended to make a distinction between medical triazoles and fungicides. If antifungal agents are considered drugs used for medical applications, it indeed is not necessary to add ‘clinical’. We have changed “clinical antifungal” to “antifungal agents” throughout the manuscript.
The MICs of the original strain (TR34/L98H (V30-40)) we used in this experiment, is listed in Table 2
Table 2: In vitro azole susceptibility profiles of TR34/L98H and TR343/L98H.
|
|
|
MIC (µg/ml) |
|
Isolate code |
Genotype |
ITR |
VOR |
POS |
V30-40 |
TR34/L98H |
>16 |
4 |
0.5 |
v284-37 |
TR343/L98H |
>16 |
>16 |
>8 |
- The subtitles such as “Heterokaryon compatibility test:” should be made more
prominent by larger font and/ or bolding? Or similar to what is was done under
“Results”.
Reply****: We have adjusted this format according the Journal stander formats.
- What was the MIC data for the strains used prior to the experiment,
they should be provided in Table 2;
Reply****: The original strain we used in this experiment is TR34/L98H (V30-40), and its azole phenotype is shown in Table 2.
- after all, the term WT is used. Which
bring us to what was the basis for the use of the 2 mcg/ml?
Reply****: 2 µg/ml represents the MIC50 of ancestor TR34/L98H (V30-40). We have added related information in L97.

Reviewer 3 Report
Comments
This manuscript describes in vitro evolution of A. fumigatus azole resistant mutant with tandem repeat, TR34 by exposure to voriconazole. After careful reading, I found there are several concerns in this study, in relation to research concept, reproducibility, experimental procedures, logicality, and data presentation.
- As referred in the text, the TR34(3) mutant has been recovered from a cystic fibrosis patient (ref. 9). This report (maybe the first identification of TR34(3) mutant) decreased novelty of the present study. In addition, the preceded study demonstrated that the TR34(3) and TR34 stains with highly similar microsatellites were isolated from an identical patient. This indicates that the TR34(3) strain must have been asexually derived from TR34 strain during infection. Taking into account these, the aim of the study (Line 80-81) has been answered. Thus the concept of the present study should be modified.
- The authors showed prevalence of TR34(3) strain during 15 cycle-experiment (Fig. 1). I think this needs biological replicates (at least 3), because the questions were when the resistant strains were emerged and how were they enriched. According to the experimental design, once the mutation was emerged in the population, the strain with reduced susceptibility to VOR can be enriched and ended in clonal in the following cycles under VOR pressure. The TR34(3) strains obtained from only one evolution experiment are not enough to convince the relation to the increased expression of cyp51A and decreased susceptibility to POS and VOR.
- In the preceded study (ref. 9), the TR34(3) strains were isolated from a patient. Curiously, MICs for POS and VOR presented in the paper were virtually comparable between the TR34(3) and TR34 strains. This indicated that duplication (2 to 3 copies) of the 34bp tandem repeat did not cause reduced resistance to the azoles. It can be also assumed that cyp51A expression was not increased in the strains. These results apparently contradict the finding in the present study. The author have to explain the gap with reasonable data.
- Genome comparison revealed 3 mutated loci other than duplication of the tandem repeat. The possibility that these mutation led to the phenotypes such as increased expression of cyp51A and changed MIC for POS and VOR was not completely ruled out.
- The aim of competition experiment (Fig. 5) was unclear. Colony growth rate in the presence and absence of VOR was examined in the former experiment (Fig. 4). Furthermore, it was not properly described how the white mutants were created. It is unclear if there were no mutations than white mutation affecting VOR susceptibility and competition with other strains.
Minor
- Line 99: What is the bottle? Please provide more precise explanations.
- Line 103: What do 1% mean? Volume? How many milliliters do you use to suspend the spore from a single bottle?
- Line 124-127: Title and explanation are redundant. Please more precise explanation.
- Line 133: How did you obtain and determine the nia and cnx mutants (mutations)? Please provide more detailed explanations.
- Line 156-161: Provide the procedure for expression analysis. If you conduct, the primer information is required. The information for the culture condition must be provided.
- Line 162-163: How did you assay spore production?
- The sequence read data should be deposited to the public database.
- There were no discussion at all for the mutations found in the evolved strain (Table S1).
Author Response
Response to Reviewer 3
This manuscript describes in vitro evolution of A. fumigatus azole resistant mutant with tandem repeat, TR34 by exposure to voriconazole. After careful reading, I found there are several concerns in this study, in relation to research concept, reproducibility, experimental procedures, logicality, and data presentation.
Reply****: Thank you for your thoughtful review. According to your comments, we have provided further information on research design and detailed methods and in-depth results.
- As referred in the text, the TR34(3) mutant has been recovered from a cystic fibrosis patient (ref. 9). This report (maybe the first identification of TR34(3) mutant) decreased novelty of the present study. In addition, the preceded study demonstrated that the TR34(3) and TR34 stains with highly similar microsatellites were isolated from an identical patient. This indicates that the TR34(3) strain must have been asexually derived from TR34 strain during infection. Taking into account these, the aim of the study (Line 80-81) has been answered. Thus the concept of the present study should be modified.
Reply****: Thank you for your comment. The case report (ref 9) indicates that selection of TR343 can take place in vivo. However, the patient case does not demonstrate that TR343 was derived through asexual reproduction. Although we agree this is likely, the problem with the patient case is that an ancestor TR34/L98H isolate was not available (see ref 9, Table 3; patient RH-5). The one TR34/L98H isolate that was analysed from this patient, was cultured after TR343/L98H had been recovered, and is likely to have been exposed to similar azole evolutionary pressure as the TR343/L98H isolates. As a consequence, we think that the aim of our study has not been answered, but that the case underscores the clinical relevance of our laboratory experiments.
- The authors showed prevalence of TR34(3) strain during 15 cycle-experiment (Fig. 1). I think this needs biological replicates (at least 3), because the questions were when the resistant strains were emerged and how were they enriched. According to the experimental design, once the mutation was emerged in the population, the strain with reduced susceptibility to VOR can be enriched and ended in clonal in the following cycles under VOR pressure. The TR34(3) strains obtained from only one evolution experiment are not enough to convince the relation to the increased expression of cyp51A and decreased susceptibility to POS and VOR.
Reply****: Our study shows a proof-of-principle. We believe that TR34 variations may be very rare as isolates harbouring TR343 have not been reported until recently, despite that TR34 has been found in environmental and clinical isolates since 1998.
In our experience experimental evolution experiments with A. fumigatus may show a wide variety of mutations and azole phenotypes. Therefore, the likelihood of reproducing our observation is low. However, we believe that reproducing this observation does not increase the validity of our observation.
- In the preceded study (ref. 9), the TR34(3) strains were isolated from a patient. Curiously, MICs for POS and VOR presented in the paper were virtually comparable between the TR34(3) and TR34 strains. This indicated that duplication (2 to 3 copies) of the 34bp tandem repeat did not cause reduced resistance to the azoles. It can be also assumed that cyp51A expression was not increased in the strains. These results apparently contradict the finding in the present study. The author have to explain the gap with reasonable data.
Reply****: In ref9, MICs for POS and VOR presented in the paper were virtually comparable between the TR343/L98H and TR34/L98H strains, while a significant change in VOR and POS phenotypes was observed in our in vitro experiments. As indicated the strains analysed from the patient case did not include an ancestor isolate, which makes any conclusions regarding temporal phenotype changes impossible. Furthermore, the conditions that the isolates were exposed to in the patient lung are uncontrolled including possible stressors not related to antifungal drug exposure. As we have shown previously (41) numerous SNP’s may be acquired in-host. As the patient isolates were not fully sequenced it remains unclear what the role of cyp51A-mutations are in these isolates.
- Genome comparison revealed 3 mutated loci other than duplication of the tandem repeat. The possibility that these mutation led to the phenotypes such as increased expression of cyp51A and changed MIC for POS and VOR was not completely ruled out.
Reply****: These three SNP’s included one in gene AFUA_2G02690 /fungal specific transcription factor, and has a homolog in yeast, ASG1, that interacts with ERGosterol biosynthesis (ERG9) (http://www.aspergillusgenome.org/cgibin/locus.pl?locus=AFUA_2G02690&organism=A_fumigatus_Af293) (36); AFUA_1G06920 Serine / threonine protein kinase which may be involved in growth (http://www.aspergillusgenome.org/cgibin/locus.pl?locus=AFUA_1G06920&organism=A_fumigatus_Af293); and AFUA_5G06450/Vacuolar protein sorting protein DigA which may be involved in vacuole organization (http://www.aspergillusgenome.org/cgi-bin/locus.pl?locus=AFUA_5G06450&organism=A_fumigatus_Af293). As these SNP’s are not associated with triazole resistance, it is likely that the TR mutations are responsible for the observed triazole phenotypes. Performing additional experiments to rule these mutations out as causal to the phenotypic changes is beyond the scope of our current paper. We have added the related results in L224-233.
- The aim of competition experiment (Fig. 5) was unclear. Colony growth rate in the presence and absence of VOR was examined in the former experiment (Fig. 4). Furthermore, it was not properly described how the white mutants were created. It is unclear if there were no mutations than white mutation affecting VOR susceptibility and competition with other strains.
Reply****: Figure 5 mimics a situation where TR343/L98H and TR34/L98H co-exist in the presence of VOR. Under VOR pressure, TR343/L98H outcompeted the ancestor TR34/L98H isolate, supporting the observed selection and dominance of TR343/L98H in our evolution experiment (Figure 4C).
We visually selected a spontaneous white color mutant from the ancestral TR34/L98H by microscopic inspection of a three days old culture grown on an MEA plate covered with a high-density of conidial sporeheads.
In order to check whether the colour has effect on VOR susceptibility and competition with other isolate, TR34/L98H with green conidia and a derived mutant with white conidia were 1:1 mixed. They showed the same fitness with equal appearance of white and green conidia at the rim of the colony in the presence and absence of VOR after 2.5 days of growth at 37℃ (Figure 5B and D). This indicates that there is no effect of the white phenotype on VOR susceptibility and competition with the green phenotype isolate.
Minor
- Line 99: What is the bottle? Please provide more precise explanations.
Reply****: The main text has been adjusted in L99.
This evolutionary line was inoculated by placing a droplet (5 µl) of asexual conidia suspension from the ancestor colony into a 30ml glass bottle (infusion bottle, VWR, Netherlands) with MEA supplemented with VOR and the bottle was incubated at 37℃.
- Line 103: What do 1% mean? Volume? How many milliliters do you use to suspend the spore from a single bottle?
Reply****: The main text has been adjusted in L101-103 as below.
The colonies grown in these bottles represented the first cycle. After seven days, A. fumigatus conidia were harvested with 5ml saline (distilled water with NaCl 0.8 g/L) supplemented with Tween 80 (0.05%), by washing off the fungal growth with beads. Asexual conidia suspensions were stored and an inoculum of 1% (50µl) of the conidia was used to initiate a next selection cycle.
- Line 124-127: Title and explanation are redundant. Please more precise explanation.
Reply****: The main text has been adjusted in L25-127.
Testing for isogeneity between ancestor and evolved isolates: In order to test whether the evolved A. fumigatus colony was isogenic with the ancestor strain, we applied heterokaryon compatibility testing, microsatellite locus genotyping, and whole genome sequencing (WGS).
- Line 133: How did you obtain and determine the nia and cnx mutants (mutations)? Please provide more detailed explanations.
Reply****: The detailed information has been provided in L132-139 and Appendix B.
Recessive markers (nitrate non-utilizing mutations nia and cnx) were introduced by ultra violet (UV 60s; 20 erg/mm2/sec) radiation in ancestor isolate TR34/L98H and any evolved genotypes. Nia and cnx mutants were isolated on basis of chlorate resistance and characterised for their ability to utilise urea, hypoxanthine, nitrite and nitrate as sole N-source (Appendix B). The isolates with complementing nia and cnx markers were co-inoculated on medium with nitrate as the sole N-source. Heterokaryons of compatible complementing strains show typical vigorous growth after 5 days, whereas inability to form heterokaryons results in spares growth of the individual homokaryons.
Appendix B
Isolation and classification of nitrate non-utilising mutants based on resistance to chlorate. For TR34/L98H strain we chose a nia (chlorate resistant, nitrate non-utilising, hypoxanthine and nitrite utilising) and for TR343/L98H a cnx (chlorate resistant, nitrate and hypoxanthine non-utilising, nitrite utilising) mutant was selected.
A: Chlorate resistant mutants appear as sectors after UV radiation 60s (20 erg/mm2 /sec).
B: Master plate of putative chlorate resistant mutants
C: Control plate with urea as N-source
D: hypoxanthine as N-source
E: Nitrite as N-source
F: Nitrate as N-source
- Line 156-161: Provide the procedure for expression analysis. If you conduct, the primer information is required. The information for the culture condition must be provided.
Reply****: The detailed information has been provided in L159-171.
The cyp51A expression was analyzed from duplicate cultures of the ancestor TR34/L98H isolate and evolved genotypes and a WT control isolate (V256-07). Strains were cultured for 16 h in 50 ml of Vogel’s MM at 37℃ at 200 rpm. Harvested mycelia were snap-frozen in liquid nitrogen and homogenized with a MagNALyser. RNA was isolated using RNAzolB (Sigma, Aldrich, Germany) according the manufacturers protocol. cDNA was synthesized using random hexamer primers and Transcriptor reverse transcriptase (Roche, Meylan, France). Real-time PCR was performed for the cyp51A and actin genes using the PCR Master kit (Roche, Meylan, France). Cyp51A and actin amplification was performed using specific primer/probe sets for each gene (Cyp51A: forward primer 5’- GTGCTCCTTGCTTCACCTG-3’, reverse primer 5’- TCCTGCTCCTTAGTAGCCTGGTT-3’, probe 5’-6Fam- AGTGACAGCCCTCAGCGACGAA-BBQ-3’; Actin: forward primer 5’- ATTGCTCCTCCTGAGCGTAAATAC-3’, reverse primer 5’- GAAGGACCGCTCTCGTCGTAC-3’, probe 5’-6Fam- TCTGGCCTCTCTGTCCACCTTCCA-BBQ-3’). Expression levels were calculated using the delta-delta-Ct method and normalized for WT expression levels [35].
- Line 162-163: How did you assay spore production?
Reply****: The detailed information has been provided in L174-179 as below:
We inoculated 5 µl of conidia suspension in a 10m glass bottle of MEA medium with and without 2μg/mL of VOR. After four days of incubation at 37 ℃, conidia were harvested with 5ml saline (distilled water with NaCl 0.8 g/L) supplemented with Tween 80 (0.05%), by washing off the fungal conidia with beads. The total conidia production was measured in three replicates by Casy® TT cell counter (OLS OMNI Life Science, Germany).
- The sequence read data should be deposited to the public database.
Reply****: The sequence data has been deposited to NCBI, and has included this information in L221. Raw sequence data are available via NCBI, BioProjectID: PRJNA666755.
- There were no discussion at all for the mutations found in the evolved strain (Table S1).
Reply****: We have added the detailed information in L224-234 as below.
These three SNP’s included one in gene AFUA_2G02690 /fungal specific transcription factor, and has a homolog in yeast, ASG1, that interacts with ERGosterol biosynthesis (ERG9) (http://www.aspergillusgenome.org/cgibin/locus.pl?locus=AFUA_2G02690&organism=A_fumigatus_Af293) (36); AFUA_1G06920 Serine / threonine protein kinase which may be involved in growth (http://www.aspergillusgenome.org/cgibin/locus.pl?locus=AFUA_1G06920&organism=A_fumigatus_Af293); and AFUA_5G06450/Vacuolar protein sorting protein DigA which may be involved in vacuole organization (http://www.aspergillusgenome.org/cgi-bin/locus.pl?locus=AFUA_5G06450&organism=A_fumigatus_Af293). As these SNP’s are not associated with triazole resistance, it is likely that the TR mutations are responsible for the observed triazole phenotypes. Performing additional experiments to rule these mutations out as causal to the phenotypic changes is beyond the scope of our current paper. We have added the related results in L224-233.

Round 2
Reviewer 3 Report
Comment to the revised version manuscript
Although the authors answered to my comment and modified the manuscript, my major concern was not properly addressed. In particular, the authors have to focus on the SNP in AFUA_2G02690. This gene has been characterized as AtrR transcription factor that plays an essential role in regulating cyp51A expression. Therefore the author’s claim that these SNPs are not likely to be associated with triazole resistance was not acceptable. Accordingly, if the mutation in atrR gene solely causes increased cyp51A expression and VOR resistance, the SNP must have appeared before TR34(3) was emerged during exposure cycles. In this case, VOR selection could result in atrR mutation but not in TR34 duplication (2 to 3). The emergence of the mutation in atrR can be clarified by sequencing the gene of strains from 5th and 10th cycle. If isolates with SNP in atrR and without TR34(3) or vice versa are included, characterization of the strain may help to prove the concept for this study’s aim.
- MIC50 -> 1/2 MIC (or Did you confirm germination rate (<50%) at the concentration of VOR?)
- L134-136: Please describe in detail what kind of N-sources the each mutant can utilize.
- Appendix B: Please indicate the strains selected as nia and cnx mutants in panels C to F by a red circle.
- Appendix A: Explanations for the data in legend are hard to understand. Please provide information for each column.
- Figure 5: Please indicate presence and absence of VOR (+VOR and -VOR) next to the panels for easy understanding.
- L307-309: As this study is a proof of concept, this sentence is wrong. Opposite. It should be : Our laboratory observations support the recent report of recovery of ~.
- SNP’s -> SNPs
